Frog eat frog: exploring variables influencing anurophagy

Measey G. John 1 john@measey.com
Vimercati Giovanni 1
de Villiers F. André 1
Mokhatla Mohlamatsane M. 1
Davies Sarah J. 1
Edwards Shelley 1
Altwegg Res 2 3
1 Centre for Invasion Biology, Department of Botany & Zoology, Stellenbosch University , Stellenbosch , South Africa
2 Statistics in Ecology, Environment and Conservation, Department of Statistical Sciences, University of Cape Town , Rondebosch, Cape Town , South Africa
3 African Climate and Development Initiative, University of Cape Town , South Africa
Hedrick Ann
Electronic publication date: 2015 Aug 25
Publication date: 2015
Volume: 3
Electronic Location ID: e1204
Received 2015 May 28; Accepted 2015 Jul 30
Copyright: © 2015 Measey et al.
Copyright year: 2015
Copyright holder: Measey et al.
License: This is an open access article distributed under the terms of the Creative Commons Attribution License, which permits unrestricted use, distribution, reproduction and adaptation in any medium and for any purpose provided that it is properly attributed. For attribution, the original author(s), title, publication source (PeerJ) and either DOI or URL of the article must be cited.
License URL: https://creativecommons.org/licenses/by/4.0/

Keywords: Anura, Cannibalism, Habitat, Invasive, Predation, Size relationships, Anurophagy

Funding: NRF ERANET BiodivERsA 2013-18 DST-NRF Centre of Excellence for Invasion Biology GJM & RA received NRF incentive funding. GJM, FAdV and SE were funded by ERANET BiodivERsA 2013-18 grant (INVAXEN). GJM, GV, FAdV, MMM, SJD and SE were funded by the DST-NRF Centre of Excellence for Invasion Biology. The funders had no role in study design, data collection and analysis, decision to publish, or preparation of the manuscript.

==============================
Background. Frogs are generalist predators of a wide range of typically small prey items. But descriptions of dietary items regularly include other anurans, such that frogs are considered to be among the most important of anuran predators. However, the only existing hypothesis for the inclusion of anurans in the diet of post-metamorphic frogs postulates that it happens more often in bigger frogs. Moreover, this hypothesis has yet to be tested.

Methods. We reviewed the literature on frog diet in order to test the size hypothesis and determine whether there are other putative explanations for anurans in the diet of post-metamorphic frogs. In addition to size, we recorded the habitat, the number of other sympatric anuran species, and whether or not the population was invasive. We controlled for taxonomic bias by including the superfamily in our analysis.

Results. Around one fifth of the 355 records included anurans as dietary items of populations studied, suggesting that frogs eating anurans is not unusual. Our data showed a clear taxonomic bias with ranids and pipids having a higher proportion of anuran prey than other superfamilies. Accounting for this taxonomic bias, we found that size in addition to being invasive, local anuran diversity, and habitat produced a model that best fitted our data. Large invasive frogs that live in forests with high anuran diversity are most likely to have a higher proportion of anurans in their diet.

Conclusions. We confirm the validity of the size hypothesis for anurophagy, but show that there are additional significant variables. The circumstances under which frogs eat frogs are likely to be complex, but our data may help to alert conservationists to the possible dangers of invading frogs entering areas with threatened anuran species.

Introduction

Amphibians are mostly dietary generalists. Most adult amphibians, whether they be frogs, salamanders or caecilians, consume a wide range of small invertebrate prey items (e.g., Wells, 2007). While the position of some amphibians in trophic webs has been shown to be important, the position and/or importance of the vast majority of species is unknown (Altig, Whiles & Taylor, 2007; Halliday, 2008). However, amphibians can achieve a surprisingly high biomass in some environments (Gibbons et al., 2006; Woolbright et al., 2006), and adult life history stages can be the most abundant vertebrate in some, particularly forested, ecosystems (Burton & Likens, 1975; Semlitsch, O’Donnell & Thompson, 2014). The aquatic larval stages often dominate small freshwater ecosystems both as important consumers of primary producing algae, and sometimes higher up the trophic food web (Colón-Gaud et al., 2009; Verburg et al., 2007). Due to their complex life histories, amphibians also link aquatic and terrestrial food webs both as predators and as prey (Kraus, Pletcher & Vonesh, 2011; McCoy, Barfield & Holt, 2009; Regester, Lips & Whiles, 2006). It follows then that as important members of the ecosystem and dietary generalists, amphibians may also be food for each other. Among such instances, frogs eating frogs are often the subject of anecdotal observations (see Toledo, Ribeiro & Haddad, 2007). However, there have been relatively few studies to show how widespread this is, and few hypotheses to suggest which frogs may be the most important predators of other anurans.

Being able to test the role of various different hypotheses on anuran predation would help clarify the importance of the role of frogs as predators of other anurans. This is of especially high importance when assessing the threats posed by invasive frogs to local populations, and predicting the effects of emerging invasive species. Perhaps of increasing relevance due to widespread amphibian declines is the need to predict the effects of loss of indigenous species to a species assemblage. Several authors have commented on the negative effects of invasive amphibians on populations of native anurans (Kats & Ferrer, 2003; Wells, 2007), and while these are sometimes multiple and complex threats (Kats & Ferrer, 2003; Kiesecker, 2003), they often include predation of native adult anurans by invasive frogs (Boelter et al., 2012). In a recent review, Bucciarelli et al. (2014) found that, along with plants and fishes, introduced amphibians are one of the groups that significantly affect indigenous amphibian populations.

Around 80% of predators are larger than their prey, and the relationship of body sizes between predators and their prey is a central tenet in food webs and their stability (Brose et al., 2006; Cohen et al., 1993). However, it appears that different habitats are not equal, with freshwater lakes and streams having higher predator–prey body-size ratios, something that could be linked to the gape limited nature of aquatic predators (Brose et al., 2006). Size differentials between amphibians and their prey appear to conform to this (Caldwell & Vitt, 1999), but the indeterminate nature of amphibian growth promotes cohort dominance through predation of con- and heterospecifics (Wakano, Kohmatsu & Yamamura, 2002; Woodward et al., 2005). Toledo, Ribeiro & Haddad (2007) suggested that the predator–prey size relationship is the most important predictor of anuran predation, such that larger species are capable of switching from being prey items to becoming predators. Similarly, Wells (2007) asserted that size is perhaps the most important predictor of adult frog predation on other anurans.

Amphibians inhabit a diverse array of habitats, and it seems likely that the availability of anurans as prey depends, in part, on the habitat that they occur in. For example, forests are complex habitats where anurans are capable of partitioning their space into three dimensions, including arboreal, leaf litter, aquatic and water-side guilds (cf. Williams & Hero, 1998). However, even within guilds it appears that multiple species are capable of dividing up resources (Caldwell & Vitt, 1999; Toft, 1985). Savannahs represent more two dimensional habitats where frogs might be more likely to encounter other anurans. Similarly, frogs may encounter more anuran prey in locations where there is more anuran diversity; either because increased diversity is likely to include heterospecifics that are small enough to be prey items, and/or as species are likely to move into the same physical space: i.e., frogs that are convenience predators eat anurans.

In this study, we aim to test the hypothesis that a frog’s body size is a predictor of the proportion of anuran prey in its diet. In addition, we seek a signal that may result from the habitat that the amphibians occur in, and whether there is a relationship between the diversity of anurans in the environment, and anuran predation. Lastly, we investigate whether or not invasive frogs are more likely to be anuran predators. In order to test our competing hypotheses, we compile a dataset of post-metamorphic anuran diet data from the literature. Our response variable is set at the population level as the proportion of the diet that was made up of anuran amphibians. We assume that there may be a taxonomic bias in the amphibian predators, based on correlations of skull shape with diet (Emerson, 1985) and therefore include superfamily in the analysis.

Materials & Methods

In order to obtain literature, we searched ISI Web of Science and Google Scholar for publications with ‘topic’ (for the former) or search terms ‘in the title of the article’ (for the latter) containing: ‘frog OR Anura’ and ‘diet’ OR ‘food’. We attempted to obtain all resulting papers and for each extracted a fixed set of data (below). Additional papers were added based on our expert knowledge, or through correspondence with authors.

Studies were included in the dataset if they contained information on diet of at least 20 post-metamorphic anurans representing a sample of a population of the same species from a single defined locality. In addition we required that a complete set of the following variables could be extracted: coordinates of study site; species name; number of individuals examined; total prey items; total anurans eaten (divided into eggs, larvae and post-metamorphic individuals). We also recorded the mean snout-vent length (SVL) if reported. If SVL data was missing, we generated a typical SVL (to the nearest mm) for the species from a variety of sources (e.g., guidebooks, Amphibiaweb). If studies reported on the diet of more than a single species, we included each species as a separate entry, provided that all information was discretely reported. In addition, we recorded the species of anuran eaten (if any), whether ingestion was reported as cannibalism alone, or mixed with other species. We defined ‘cannibalism’ as evidence that a frog had eaten a conspecific, irrespective of the life-history stage: i.e., egg, tadpole or post-metamorphic individual of the same species. We also recorded if the species studied was considered invasive (or not) at the site where the study was conducted, this was then later corroborated with the Global Amphibian Assessment (GAA) database (see below).

Taxonomy

We used current taxonomic nomenclature according to Frost (2015). We updated taxonomic entities when possible and checked validity of identification with distribution data (see below). In cases where the published species identification could not be verified, we removed the record from the dataset. To test for the influence of taxonomy on anuran predation, we grouped families into superfamilies, because some families were represented only by a single observation. We used a recent phylogeny of all Amphibia to group families at well supported nodes (Pyron, 2014; Pyron & Wiens, 2011): Discoglossoidea, Hyloidea, Pelobatoidea, Pipoidea, Leiopelmatoidea and Ranoidea (see Pyron, 2014 for contents of superfamilies).

Habitat

For each species in our dataset, we assigned a habitat value based on that reported in the IUCN database. We followed Ficetola et al. (2015) in assigning species to one of four habitat categories: forest specialists, grassland, shrubland and generalists. Species were scored as generalists if their entry in the IUCN database mentioned that they could be found in more than one habitat category.

Number of anuran species at the study site

To estimate anuran species diversity at each study site, we took the locality record of the study site, converted it to decimal degrees to create a layer of all the studies for which we had locality data, using ArcMap GIS 10.2. This layer was then overlaid with the anuran species range maps obtained from the IUCN Global Amphibian Assessment dataset (http://www.iucnedlist.org/amphibians; GAA version 3.1), using the extent of occurrence shapefiles as “known distributions”. After noting the number of species at each site, we cross-checked coordinates given in manuscripts with species studied or mentioned within to ascertain a correct locality and accurate species identification, and these were reconciled with the GAA list. Occasionally, coordinates did not correspond with GAA presence, and these were double-checked and if locality or species identification could not be verified with reference to the GAA list, the record was removed from the dataset.

Analyses

From each record, we calculated the total number of ‘anurans eaten’ (eggs + larvae + post-metamorphic individuals), the number of prey eaten that were ‘not anurans’ (total prey − anurans eaten). These were combined to make our response variable: (‘anurans eaten’, ‘not anurans’). We analysed these data using generalised linear models (GLM) with binomial errors and a logit link function implemented in the function ‘glm’ in R 3.2.1 (R Core Team, 2015) using our response variable, and habitat, invasiveness, SVL and species diversity as explanatory variables. None of the explanatory variables were strongly correlated with each other (r < 0.4). In a preliminary analysis, we established that superfamily was a significant effect, so we included this term in each model. We also assessed whether the number of individuals in the study had a positive relationship with finding anurans in the diet.

We fitted 15 models to the data, representing all possible combinations of the four explanatory variables of interest. We had no prior reason for thinking that there may be interactions between these variables and therefore only fitted main effects. We ranked the models using Akaike’s Information Criterion (AIC). The coefficients in the models we used are estimated as the logarithm of the odds ratio. We report these and also the odds ratios.

Results

Our literature search yielded 1,308 items, and from these we were able to find 323 papers in which all of our variables could be sourced. Since some studies reported data on multiple species, we had a dataset of 355 records from 228 species. These included the contents of 40,238 anuran stomachs with some 456,146 prey items, of which 1,711 were anuran eggs (29%), larvae (21%) and post-metamorphs (50%). The majority of records revealed that most populations of frogs did not consume any anuran life-history stages (n = 278). However, over a fifth of records (n = 77) reported predation of eggs, larvae or post-metamorphic individuals, indicating that Anura are not a rare or unusual dietary item in frog populations. However, when frogs did eat other anurans, the mean proportion of amphibian prey was only 2.9% of total prey items (max. 18.5%).

Our initial analyses revealed a significant effect of superfamily on predation of anurans, whereby members of Ranoidea and Pipoidea were significantly more likely to eat anurans than were the other superfamilies (Fig. 1, Likelihood Ratio Test: chi-square = 1,387, df = 4, P < 0.001). To account for this in subsequent analyses we included superfamily as a fixed effect in each GLM. It is noteworthy that when only post-metamorphic prey items were included, the significant effect of the pipids fell away leaving ranids as the only significant superfamily (results not shown). However, we found no increase in detection of anurophagy with increasing number of individuals in each study (P = 0.137), this suggests that we were successful in avoiding sampling bias by restricting studies to those with 20 individuals or more.

Figure 1 Taxonomic bias in anurophagy across superfamilies (and Hyloidea).

The proportion of frogs eaten in the diet of other frogs divided up by superfamily (n values are given below superfamily names). The box-plot shows the significant increase in anurans in the diets of pipids and ranid frogs. Individual data points are added (with a jitter effect to prevent overlapping), and show the range of data in all groups. (b) Inset: Data for the superfamily Hyloidea, broken down to show the effects of families (n values are given below family names). The families Ceratophryidae, Leptodactylidae, and Hylidae have large proportions while smaller proportions are eaten by Bufonidae (the single outlier is a study on invasive Rhinella marina) and Dendrobatidae.

The most parsimonious model included all of our variables: body size, habitat type, invasive species and diversity at the study site (Model ‘spp + habitat + invasive + SVL’, Table 1). A single model was within four delta AIC units of the best model (Model ‘SVL + spp + invasive’). The proportion of anurans eaten was significantly influenced by size (effect on the logit scale: 0.028, se = 0.001; odds ratio = 1.028, i.e., for every mm increase in SVL, the species is 2.8% more likely to eat other anurans), invasive species (effect of being invasive 0.336, se = 0.075; odds ratio = 1.400, i.e., invasive frogs were 40% more likely to eat anurans), the number of species at the study site (0.017, se = 0.001; odds ratio = 1.017, i.e., for every additional species occurring at the study site, frogs were 1.7% more likely to eat anurans), and habitat (Fig. 2: this effect was mainly driven by frog generalists being more likely to eat anurans than frogs from forest habitat, with frogs from forest, grassland and shrubland similar to one another: grassland vs forest −0.029, se = 0.188, odds ratio = 0.971, i.e., grassland frogs were 2.9% less likely to eat anurans than forest species).

Figure 2 Habitat influence on anurophagy.

The proportion of frogs eaten in the diet of other frogs divided by predator habitat. The box plot shows the significantly higher incidence of anurans in the diets of generalists over frogs from other habitats (n values are given below habitat categories). Individual data points are added (with a jitter effect to prevent overlapping), and show the range of data in all groups.

Table 1 General Linear Models exploring influences of anurophagy.

Generalised Linear Models with binomial errors and a logit link function run on proportion of frogs eaten (eggs + tadpoles + post-metamorphic individuals and total non-anuran prey: ‘prop’) in relation to whether populations were ‘invasive’, the ‘habitat’ the species is found in, the size of the species (‘SVL’), and the diversity of species at the site where the study was conducted (‘spp’: see methods for details of variable calculations). Each model was run with ‘superfamily’ as a fixed effect. Δ AIC is the difference in Akaike Information Criterion values (AIC) between the current model and the best, wi is the relative support a model has from the data compared to the other models in the set: Akaike weight. K is the number of parameters in the model.

Model number	Model name	log-likelihood	K	Δ(AIC)	wi	
4	spp	−2,444.33	6	1,502.096	0	
2	habitat	−2,369.12	8	1,355.662	2.93E-295	
6	habitat + spp	−2,320.74	9	1,260.91	1.10E-274	
1	invasive	−2,170.55	6	954.5391	3.71E-208	
10	invasive + habitat	−2,138.69	9	896.8023	1.28E-195	
5	invasive + spp	−2,081.61	7	778.658	5.78E-170	
13	spp + invasive + habitat	−2,025.5	10	672.4358	6.73E-147	
3	SVL	−1,793.04	6	199.5074	3.33E-44	
7	SVL + invasive	−1,791.21	7	197.8459	7.65E-44	
9	SVL + habitat	−1,780.9	9	181.2266	3.11E-40	
12	SVL + invasive + habitat	−1,777.85	10	177.1206	2.42E-39	
8	SVL + spp	−1,702.88	7	21.19062	1.75E-05	
14	SVL + spp + habitat	−1,698.55	10	18.52097	6.66E-05	
11	SVL + spp + invasive	−1,692.14	8	1.700938	0.299309	
15	SVL + spp + invasive + habitat	−1,688.28	11	0	0.700607	

In addition, we considered whether cannibalism was a major factor in studies where frogs were found eating other anurans. Cannibalism was identified in 28 of the records that we analysed. Another 28 could have been examples of cannibalism, but the identity of the anuran eaten was not specified. We thus classified interactions as cannibalistic, possibly cannibalistic and not cannibalistic. In those studies where frogs ate other anurans, cannibals were not found to have a greater proportion of anurans in their diet (Likelihood Ratio Test: Chi-square = 0.0014; df = 2,74; P = 0.3985).

Discussion

Our data show that size is a dominant predictor of anurophagy in populations of post-metamorphic frogs, confirming existing hypotheses to this effect. However, the best model fitting our data included all three additional variables to best explain the presence of anurans in the diet of other anurans; frogs eating frogs. We found that a model including size, habitat, invasiveness and the diversity of anurans at the study site was the best model. We discuss these four explanatory variables separately first, before considering how they may act together.

Body size

Although it seems obvious that larger species are more likely to consume other anurans, and our results are in accordance with this, we found a number of studies that did not fit this model. For example, very small frogs are capable of consuming the eggs of other species (Beard, 2007; Drewes & Altig, 1996). In addition, juveniles of species that attain a large adult size can eat a large proportion of anurans. For example, Glorioso et al. (2012) found that nearly all size classes of the Cuban treefrog (Osteopilus septentrionalis) ate other anurans, from under 40 mm to 83 mm. Similarly, Conradie et al. (2010) found that juveniles of Pyxicephalus adspersus had eaten many other small anurans, even though their own mean body size was under 40 mm (see also Schalk et al., 2014). This species has a very large gape, and it has been suggested elsewhere (Boelter et al., 2012; Emerson, 1985; Konopik, Linsenmair & Grafe, 2014; Schalk & Fitzgerald, 2015) that gape size may be as important as overall size in consumption of anurans. We suggest that this will be a fruitful area for future research on anurophagy. We also found a number of very large frogs which had no anurans in their diet, many of which were toads (see below).

Species diversity

We found that the number of species present at the site in which the dietary study was conducted had a positive relationship with the proportion of anurans in the diet. This result is somewhat intuitive, and might be best explained by increasing diversity resulting in increased numbers of anurans which are of a relevant size, and/or acting as a proxy for an increased abundance of amphibians in the ecosystem. Due to the nature of the data, we were not able to investigate the density of amphibians at the time that they were sampled. We expect that density of prey species would almost certainly play a role (see Polis & Myers, 1985), but this was very rarely, if ever, reported in the dietary papers that we studied. Amphibian diversity increases markedly in the tropics (Bonetti & Wiens, 2014; Hof et al., 2011), often in forested areas, and it may be that our finding correlates with a number of other untested variables including an increase in ambient temperature, or reduced seasonality. Each of these may provide interesting variables to consider. However, anuran diversity was the best variable to consider within the constraints of the available data, as it provides a broad measure of the encounter rate of one anuran with another, and hence predation opportunity.

Habitat

The habitat type in which each species occurred demonstrates the importance of generalists over all but shrubland species as amphibian predators, despite the fact that more species were found in forest than in any other habitat type. Forest habitats are physically complex and three dimensional, such that while there may be many species present, there may also be significant spatial separation (Williams & Hero, 1998), as well as escape opportunities and refugia available to the prey. However, generalist species appear to prey on frogs both in forests and in the other habitat categories.

Konopik, Linsenmair & Grafe (2014) documented an invasion of forested habitat by generalist anurans in Borneo which is particularly insightful. Following road construction, several species not known to inhabit the forest quickly moved in with one newcomer found to consume significantly more anurans than its resident congener. Interestingly, the authors found that these frogs were matched for body size, but that the newcomer had a significantly larger gape (see above).

Invasiveness

Invasive frogs are generally claimed to have a negative impact on native species through a number of mechanisms including, competition, hybridisation and predation (Bucciarelli et al., 2014; Kraus, Pletcher & Vonesh, 2011). The significance of predation has rarely been quantified, but is often listed as a threat to indigenous anurans. Our finding that invasive frogs have a higher proportion of anurans in their diet gives substance to this claim. Surprisingly, invasiveness was included in the best model in our study despite the small number of studies of the diets of invasive anurans that were available. Also, it was not confounded by the significant taxonomic signal in anuran predation, especially as our data included many studies on Rhinella marina, a bufonid which appears to rarely eat other anurans (see below). These results suggest that (non-bufonid) invasive frog species should be considered as an important threat in terms of their direct predatory effects on other anurans.

Taxonomy

Emerson (1985) predicted that vertebrate-eating frogs would have a large gape (a wide skull with long jaws) and a high jaw closing force (longer jaws posterior to adductor insertion). Her data, based on a literature review, suggest a strong taxonomic bias to prey type, and she emphasised the relationship with skull morphology. Toledo, Ribeiro & Haddad (2007) described anuran predators (including post-metamorphic anurans) as ‘convenience predators’, adding that “...the most representative predators are those who exhibit similar habits to the anurans, facilitating their encounters.” Encounters would be expected to be higher amongst pond or stream dwelling species than among those which only visit these habitats for reproduction, such as many large bufonids. In a post hoc analysis within the superfamily Hyloidea we confirm that the Bufonidae are negligible predators of anurans while the families Ceratophryidae, Hylidae and Leptodactylidae all have elevated levels of anurophagy (Fig. 1B). Indeed, the Ceratophryidae appear to regularly ingest anurans as well as other vertebrates (Schalk et al., 2014).

Our post hoc tests suggest that bufonids and dendrobatids might be unusual families within the Hyloidea. Caldwell (1996) discussed the evolution of myrmecophagy in dendrobatids, and Isacch & Barg (2002) investigated whether bufonids are also specialised ant-feeders, confirming that they tend to have smaller prey sizes than sympatric anurans in Argentina. Diesmos, Diesmos & Brown (2008) reviewed impacts of invasive frogs in the Philippines and highlighted the differences between the anuran eating behaviour of the ranid, Hoplobatrachus rugulosus compared to a largely invertebrate diet in R. marina. Undoubtedly, R. marina can and does consume other anurans (Lever, 2001; Wells, 2007; Pizzatto & Shine, 2008), but it appears that this and other toad species do not have a large proportion of anurans in their diets.

Synergy of habitat, species diversity, invasiveness and size

Our model suggests that our chosen variables are synergistic, with the combination of them being more likely to result in anuran predation by frogs. This suggests that previous workers who considered size or taxonomy alone had not appreciated that the conditions under which anuran predation by frogs occurs is considerably more complex, and also involves variables that describe the animals’ habitat and opportunities available to prey on other anurans. A few studies have provided insight into this complexity: Studies on predation by Lithobates catesbeianus found that it commonly consumed many other anurans, but not necessarily in proportion to their abundance (Boelter et al., 2012; Werner, Wellborn & McPeek, 1995), implying that this species is a habitat-specific convenience predator. In their study, Boelter et al. (2012) found that relative size was important, although all sizes of bullfrogs consumed anurans: that is, both gape and size appear important. In addition, these authors noted the low incidence of cannibalism, and ascribed it to the higher diversity of sympatric species; equivalent to our species diversity variable. However, the explanatory variables we could include in our study were tightly constrained by the need for uniform availability across the entire dataset (see below). Therefore we acknowledge that body size, species diversity, habitat and invasiveness may represent proxies for more biologically meaningful variables that we were not able to score.

Cannibalism

Some anurans are known for cannibalism by adults on earlier life-history stages; heterocannibalism (see Polis & Myers, 1985 for a review). In our data, eggs featured less prominently in heterocannibalism than did tadpoles, but this may be due to a combination of accessibility and the difference in period that each life-history stage is available. For example, frogs in the genus Xenopus are often confined in water bodies with conspecific eggs and tadpoles, and at times when these are abundant, they make up a large proportion of the prey eaten (Measey, 1998). Generally, we found pipids to be notable by their high levels of predation on eggs and tadpoles, although this is not confined to conspecifics. Pipids lack a tongue and rely on suction for most small prey items, although they are also able to take large prey through jaw prehension (Carreño & Nishikawa, 2010).

Egg predation is not confined to aquatic anurans, frogs in the genus Afrixalus are associated with egg predation on conspecific as well as heterospecific eggs (Drewes & Altig, 1996). Again, levels of predation can become significant when this prey type is in abundance (Dayton & Fitzgerald, 2006; Marshall, Doyle & Kaplan, 1990; Vonesh, 2005). In our dataset, Pelophylax ridibundus, Eleutherodactylus coqui and E. planirostris were also reported to cannibalise brood (Beard, 2007; Olson & Beard, 2012; Ruchin & Ryzhov, 2002). Brood sites for Eleutherodactylus species are in fallen leaves that capture small quantities of water, allowing other adults to easily access and cannibalise them. Drewes & Altig (1996) provide a number of anecdotal accounts of adult frogs consuming eggs, and we agree with their conclusion that this may be far more common than reported, although temporally limited by the transient availability of this prey type.

Terrestrial anuran species predated on tadpoles far more commonly that would be expected for species that rely on lingual protraction for prey capture (O’Reilly, Deban & Nishikawa, 2002). Given that this feeding mode is not effective for most aquatic prey, it appears that jaw prehension is not an unusual mechanism for many post-metamorphic frogs. Again, it is noteworthy that the majority of cases occur in the ranids (with some instances of leptodactylids eating tadpoles), and that these made up a significant proportion of prey. In one study on the diet of the African Tiger Frog, Hoplobatrachus occipitalis, the authors found that many of the prey items had an aquatic origin with fish making up the largest proportion of prey items as well as large numbers of tadpoles (Hirschfeld & Roedel, 2011). Our data suggest that aquatic ingestion, presumably by jaw prehension, is not uncommon in frogs, and we suggest that this requires further study.

In the literature we surveyed, the life-history stage most commonly eaten by frogs was post-metamorphic individuals. There are reports that eating anurans that are large in relation to the predatory frog may result in excessive handling time, as well as a risk of injury to the individual predator (Wyatt & Forys, 2004). This corroborates our finding that body size is a strong predictor of anurophagy. Unfortunately, we could not test for size differences between frogs consuming and being consumed, as these data were recorded in only a handful of studies. However other studies have found larger individuals are significantly more likely to eat smaller ones (Toledo, Ribeiro & Haddad, 2007).

Conservation management

Our study provides important insight into amphibian conservation management with respect to direct predation effects. Clearly, there is the important threat from invasive species, which our study suggests has taxonomic bias. In a post hoc test excluding all records of members of the family Bufonidae, the effect of invasive species increased to invasive frogs being 53.3% more likely to consume anurans (full glm model with all variables se = 0.08). Managers might also expect that the behaviour of large frogs may change if the habitat is disturbed, especially in situations where amphibian diversity is high. Furthermore, our traits may be useful in scores on risk assessments when considering which amphibians should not be traded. Many countries are drawing up lists of species that should not be traded, and these are largely based on species that are already invasive elsewhere. The results of our study suggest that avoiding trade in large ranid (and pipid) generalists would appear to be prudent, and that this may be especially important for countries with high amphibian diversity.

Constraints

There were a large number of papers that we could not include in our study due to the lack of uniform variables. Most commonly, investigators reported on statistically derived figures representing prey proportions or indices, without inclusion of, or any way to calculate back to, the raw data. The importance of including raw data in manuscripts (or at least in supplementary information) is invaluable in making such data available for additional studies (such as this one, but see also Vignoli & Luiselli, 2012).

Our study probably underestimates the importance of anurophagy when it occurs. For example, we did not consider any anurophagy that takes place before metamorphosis, although this is recognised as being of great importance, especially with respect to cannibalism (see Altig, Whiles & Taylor, 2007; Polis & Myers, 1985). Further, we chose to represent the importance of anurans in the diet of frogs as a proportion of total prey items ingested, because data on prey volume was missing from the majority of studies. The proportion of total prey items provides no clear indication of what volume anurans made up in the stomachs of individuals or even at a population level. Some authors have addressed the volumetric contribution or contribution by mass of different prey items, and these generally increase the importance of large items, such as frogs (e.g., Boelter et al., 2012). Lastly, the data available to us in the dietary studies we used rarely quantified behavioural traits of frog predators such as activity patterns and foraging mode. These likely remain important and largely underexplored topics with respect to anurophagy (see Schalk & Fitzgerald, 2015).

Conclusions

We find that the consumption of anurans by post-metamorphic frogs is not unusual at the population level, but does have a strong taxonomic bias. A high proportion of frogs in the diet is predicted for animals that are large, invasive, generalists, and live in areas with many other amphibian species. Our study was limited by the type of available data, and future investigations may need to focus on traits such as gape, density and predator behaviour to better refine incidents of anurophagy. The limited number of studies on invasive amphibian diet suggests that they commonly eat other anurans, and it seems reasonable to assume that invasive pipids and ranids may be expected to have an adverse effect on local amphibian communities. This should be taken into account in pre-border risk assessments of these taxa. Overall we found a small number of papers available on invasive anuran diets, and we specifically encourage more studies on the impacts of invasive anurans.

Supplemental Information

Supplemental Information 1 Literature on frog diet detailing variables used in the study

Literature references for data used in the study. Variables extracted are: coordinates of study site; Country of study; Superfamily; family; genus and species name; whether or not the species was invasive at the study site; habitat; snout-vent length (SVL); whether there was evidence of cannibalism; the richness of amphibians at the study locality; number of individual frogs examined; proportion of anurans in diet.

Click here for additional data file.

We would like to thank the authors who kindly provided us with data not originally published with their studies. In addition, we’d like to thank Solveig Vogt, Marie Theron and James Vonesh for their help with literature and critical reading of the manuscript. Two reviewers gave insightful comments that improved the manuscript.

Additional Information and Declarations

Competing Interests

Author Contributions

John Measey is an Academic Editor for PeerJ.

G. John Measey conceived and designed the experiments, performed the experiments, analyzed the data, wrote the paper, prepared figures and/or tables.

Giovanni Vimercati, F. André de Villiers, Mohlamatsane M. Mokhatla, Sarah J. Davies and Shelley Edwards conceived and designed the experiments, performed the experiments, wrote the paper.

Res Altwegg analyzed the data, wrote the paper.

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
