# Peer review of "Frog eat frog: exploring variables influencing anurophagy"

_PeerJ, doi:10.7717/peerj.1204_

## Round 0.1 · original submission · Major Revisions

Please revise paying special attention to the more critical review. Note that both reviewers asked for more clarity in the writing.

·

Basic reporting

See general comments below.

Experimental design

No major issues

Validity of the findings

No Comments

Additional comments

This is an interesting analysis of the factors that might explain why frogs eat other frogs. I have no major issues with the analyses done or the broad conclusions drawn. Clearly there is more work to do on this issue, but perhaps this paper can provide some clues for how to start.

My only general comment is that there are some problems with the writing in places. The paper is often not very clearly written. This especially becomes a problem when it is difficult for the reader to determine whether the authors are referring to frogs as predators or frogs as prey (since the manuscript deals with both). This needs to be perfectly clear in every case, and this should be possible without interrupting the flow of the article. In general some more attention to sentence structure and clarity would greatly improve the readability of this manuscript.

Specific comments:

Line 85: The Wells book was published in 2007.

Line 90-94: I’m not sure this comparison between forests and savannas make sense. The relevant variable is the number of species (or individuals?) per unit area. Just because there are more potential guilds, and more complexity, within a forest, doesn’t mean that species are less likely to encounter one another there. Habitat complexity may promote amphibian diversity, which by the mechanism mentioned by the authors later in the paragraph, could increase the likelihood of anuran on anuran predation.

Line 164: Are eggs as easily detectable as other types of prey (i.e. post-metamorphs) in stomach content analyses? Might this bias the results somewhat against detection of batracophagy in smaller anurans, which could probably only eat eggs?

Line 167: The variables ‘eaten’ and ‘not eaten’ need to be defined. The naming of the variables almost makes it sound like the authors are looking at whether a given species was eaten by other amphibians, but I think it’s the other way around: which species eat amphibians. In any event, these should be explained better; perhaps in the sentence starting on line 162.

Line 175: I don’t think it’s a big deal, but there are some plausible interactions. For instance, I could imagine that the effect of invasiveness could be influenced by body size. I understand, however, that the authors may wish to keep the number of models to a reasonable number.

Line 186: Is it possible with your dataset to give the percentage occurrence for each of these life-history stages? i.e. of the anurans eaten, how many were in the egg stage vs. in the adult stage?

Line 188: This could be worded better to clarify that what is not in numerical abundance is the anurans as a component of the diet (as opposed to the frogs that are eating them; this wasn’t clear).

Line 197: The description of this analysis on line 171 seems to imply that you are interested in a binary measure of whether or not any amphibian predation was detected based on sample size. Here, it seems that you are testing instead whether sample size affected the measurement of the proportion of amphibians eaten. I think the former is a better test of sampling bias, and would suggest something like a logistic regression to test it (response variables being 0=no amphibian predation detected; 1=amphibian predation detected).

Line 195: Be clear that you mean post-metamorphic prey items. This is a difficult task since they are both coming from the same group of animals, but it is essential that you point out when you are talking about predators and when you are talking about prey.

Line 211: Wording is confusing. I suggest “generalists being more likely to eat anurans than frogs from forest habitat, with frogs from forest, grassland and shrubs similar to one another”

Line 249: “but this was”

Line 266: “insightful invasion”?

Line 300: awkward wording, use “are” instead of “aren’t”

Line 309-310: This statement raises the question of whether interactions between these variables are indeed important. Might it not be worth exploring a few such models?

Line 318: What is this “(gape and size)” supposed to mean? Explain, or remove.

Line 336: An awkward sentence, should be rephrased.

Line 378: The use of ‘batracophagy’ throughout may help clear up some of the confusing writing.

Line 380: what result, specifically? In which direction was the correlation?

Table 1: The authors may want to consider ordering the models in terms of ascending or descending AIC score. As currently written, it is difficult to compare the models in terms of their fit to the predictor.

Line 555: Why are the Bufonids and Dendrobatids specifically highlighted (both here and in main text)? The inset makes it seem as though most of the other families also don’t eat anurans.

Reviewer 2 ·

Basic reporting

The title is very uninformative as it does not accurately describe the study. It needs to be revised/changed.
The introduction was rather unorganized and the writing lacked clarity. There were several general statements made without a supporting reference. Also, the first paragraph made an awkward transition in introducing the purpose of the study.
There were several instances where literature was not appropriately referenced (noted in the track changes). Also, there were some useful references that were not discussed which would contribute to the discussion and provide a better context to understand the anurophagy:
Scott & Aquino 2005. It’s a frog-eat-frog world in the Paraguayan Chaco: food habits, anatomy, and behavior of the frog-eating anurans. Pp. 243–259, in Ecology and Evolution in the Tropics: A Herpetological Perspective.
Schalk & Fitzgerald. 2015.Ontogenetic shifts in ambush site selection of a sit-and-wait predator, the Chacoan Horned Frog (Ceratophrys cranwelli).Canadian Journal of Zoology. 93:461-467.
Schalk et al. 2014.On the diet of the frogs of the Ceratophryidae: Synopsis and new contributions. South American Journal of Herpetology. 9:90-105.
In the discussion, there should be another paragraph added that briefly discusses the other important factors that would influence anurophagy. This should include difference (or similarity) in activity patterns, ontogenetic niche shifts, and foraging modes. All these factors are going to affect encounter rates among conspecifics and heterospecifics and should be discussed as they are a large limitation to the generality of the results of the study.
I found Figure 1 difficult to interpret. I think the insets should be removed and made into their own separate figures (see my comments in track changes).

Experimental design

The hypotheses need to be explicitly stated and clarified in this last paragraph because as it is written now, they are not clear.
Why was a sample size of 20 individuals established as a cutoff? There are numerous of observations available in Herpetological Review that are going to be overlooked as a result. In addition, many of the larger frogs may be less numerically abundant and result in smaller sample sizes.
The articles used in the meta-analysis need to be provided as an appendix for the reader.

Validity of the findings

Could the taxonomic bias simply be due to the criteria established for the study? There are many instances of anurophagy documented in Herpetological Review natural history notes that would be overlooked due to the criteria of the study.

There is a problem with discussing the results in terms of conservation; only interactions with the post-metamorphic lifestage were examined in the context of this study. While there may be a taxonomic bias at the post-metamorphic lifestage, what would happen is this included the larval stage? There have been numerous studies documenting the impact of invasive larval Cane Toads on native amphibians. This, coupled, with the fact that new data is emerging on the trophic status of tadpoles to suggest a greater degree of omivory and carnivory (see Altig et al. 2007, which is cited in this ms) could have large implications for conservation management, yet this is not even acknowledged or discussed in the constraints section of the manuscript.

Additional comments

The authors attempt to document the occurrence and prevalence of anurophagy in anuran amphibians. I found the study to be interesting and the analyses appropriate as the results could have implications for conservation biologists and ecologists.
I think the criteria established of suitable studies to include by the study greatly limited the sample size of the studies used. The minimum sample size of 20 individuals likely excluded a number of suitable papers which likely lead to the taxonomic bias observed in their results. This in turn, lead to the broad scope of the variables included in the analysis (e.g. use of superfamilies rather than families) and not including other intrinsic traits of the species in the dataset which are important predictors of whether anurophagy would occur (e.g., activity patterns, foraging mode).
I have made numerous comments and edits on the MS word document using track changes. I hope these will benefit the authors and lead to a much improved manuscript.

Annotated reviews are not available for download in order to protect the identity of reviewers who chose to remain anonymous.

---

## Round 0.2 · accepted · Accept

Thank you for your changes.